# Nonlinear molecular dynamics of quercetin in *Gynocardia odorata* and *Diospyros malabarica* fruits: Its mechanistic role in hepatoprotection

Arabinda Ghosh[1], Pranjal Sarmah[2], Harun Patel[3], Nobendu Mukerjee[4], Rajbardhan Mishra[5], Saad Alkahtani[6]*, Rajender S. Varma[7], Debabrat Baishya[2]*

1 Microbiology Division, Department of Botany, Gauhati University, Guwahati, Assam, India, 2 Department of Bioengineering and Technology, GUIST, Gauhati University, Guwahati, Assam, India, 3 R. C. Patel Institute of Pharmaceutical Education and Research, Shirpur, Maharastra, India, 4 Department of Microbiology; Ramakrishna Mission Vivekananda Centenary College, Khardaha, West Bengal, Kolkata, India, 5 Laboratory of Immunotherapy, Institute of Microbiology v.v.i., Czech Academy of Sciences, Videnska, Prague, Czech Republic, 6 Department of Zoology, College of Science, King Saud University, Riyadh, Saudi Arabia, 7 Regional Center of Advanced Technologies and Materials, Czech Advanced Technology and Research Institute, Palacky University, Olomouc, Czech Republic

☯ These authors contributed equally to this work.
* drdbaishya@gmail.com (DB); salkahtani@ksu.edu.sa (SA)

**Data Availability Statement:** All relevant data are within the manuscript and its Supporting Information files.

## Abstract

Liver performs number of critical physiological functions in human system. Intoxication of liver leads to accumulation of free radicals that eventually cause damage, fibrosis, cirrhosis and cancer. Carbon tetrachloride ($CCl_4$) belongs to hepatotoxin is converted to a highly reactive free radical by cytochrome P450 enzymes that causes liver damage. Plant extracts derived quercetin has substantial role in hepatoprotection. This study highlights the possible mechanism by which quercetin plays significant role in hepatoprotection. HPLC analysis revealed the abundance of quercetin in the fruit extracts of *Gynocardia odorata* and *Diospyros malabarica*, were isolated, purified and subjected to liver function analysis on Wistar rats. Post quercetin treatment improved liver function parameters in the hepatotoxic Wistar rats by augmenting bilirubin content, SGOT and SGPT activity. Gene expression profile of quercetin treated rats revealed down regulation of HGF, TIMP1 and MMP2 expressed during $CCl_4$ induction. *In silico* molecular mechanism prediction suggested that quercetin has a high affinity for cell signaling pathway proteins BCL-2, JAK2 and Cytochrome P450 Cyp2E1, which all play a significant role in $CCl_4$ induced hepatotoxicity. In silico molecular docking and molecular dynamics simulation have shown that quercetin has a plausible affinity for major signaling proteins in liver. MMGBSA studies have revealed high binding of quercetin ($\Delta G$) -41.48±11.02, -43.53±6.55 and -39.89±5.78 kcal/mol, with BCL-2, JAK2 and Cyp2E1, respectively which led to better stability of the quercetin bound protein complexes. Therefore, quercetin can act as potent inhibitor against $CCl_4$ induced hepatic injury by regulating BCL-2, JAK2 and Cyp2E1.

**Funding:** This work is funded by Researchers Supporting Project (RSP- 2021/26), King Saud University, Riyadh, Saudi Arabia.

**Competing interests:** The authors have declared that no competing interests exist.

## Introduction

Liver plays a crucial role in the detoxification of xenobiotic compounds, toxins, and chemotherapeutic drugs alongside its usual function of metabolism, secretion, and storage. Carbon tetrachloride ($CCl_4$), a potent hepatotoxin, is activated by the enzyme cytochrome P450 in endoplasmic reticulum. It causes the synthesis of highly reactive free radical products followed by lipid peroxidation in liver [1]. Hepatic tissue injury is instigated by those free radical derivatives mediated by lipid peroxidation [1]. The liver microsomal cytochrome P450 interceded lipid peroxidation and tissue damage by trichloromethyl free radicals are well established facts [2–4]. In addition, many cell signal-transducing proteins as well as major transcription activator elements e.g., SATA3, JAK2 (Janus Kinase 2), and BCL-2 (B-cell lymphoma-2) etc. are upregulated by $CCl_4$ resulting in hepatotoxic effects [5]. $CCl_4$-induced hepatotoxicity, on the other hand, exhibited apoptotic response declination by increasing BCL-2 (B-cell lymphoma-2) related putative X protein and downregulating apoptotic regulator BCL-2; thus, BCL-2 was demonstrated to be a key role on regulatory pathway in cancer development [6].

Medicinal plants have traditionally been used to cure several diseases, including liver disorders, without causing much toxic side effects. Numerous plants have been investigated hitherto against liver toxicity ailments. A few that have extensively been used includes *Camellia sinensis*, *Picrorhiza kurroa*, *Glycyrrhiza glabra*, *Silybum marianum* and *Curcuma longa* etc. [7–9]. The antioxidants present in medicinal plants may cure different diseases by offering cytoprotection from damage caused by free radicals, the highly reactive oxygen compounds [6]. Analogous to Braviscarpin's crude flavonoid, natural flavonoids have been widely employed for the treatment of hepatotoxicity, as in the case of reversing the change in JAK2, BCL-2, and SATA3 [6]. A polymethoxy flavonoid has recently been reported to have anti-inflammatory and immunomodulatory action in enhancing BCL-2 to lower the effect of $CCL_4$-induced hepatotoxicity [10]. *Silybum marianum* derivedcompound, silymarin, has profound antioxidant and hepatoprotective activity as it inhibits the free radical induced liver toxicity produced from $CCl_4$, acetaminophen and ethanol [11]. In another report, Mahli and coworkers investigated the potent hepatoprotective activity of silymarin in rats having acute liver injury [1].

Quercetin, an important dietary flavonoid has a wide range of health benefits, including antioxidative, anti-inflammatory, and anti-apoptotic attributes [12]. Quercetin has been shown to present in many plants and this flavonoid has been shown to protect cells against oxidative stress caused by xenobiotics [13, 14]. Furthermore, quercetin has been shown to protect liver from hepatotoxin-induced damage [15]. Due to its antioxidant properties in addition to hepatoprotective action, bark extract of *Diospyros malabarica* (Ders.) Kostel has been reported to possess a wide range of therapeutic applications. [16] The seed extracts of *Gynocardia odorata* R. Br. hasbeen used traditionally as anti-diabetic and antiulcer agent. For instance, phytoconstituents from methanolic extract of *Gynocardia odorata* R.Br. were reported to encompass many antidiabetic, anti-inflammatory and hepatoprotective compounds [17]. *Diospyros malabarica* and *Gynocardia odorata* has been reported to contain quercetin [18]. Owing to high importance and usefulness of these two medicinal plants, the present investigation has centered on the evaluation of quercetin responsible for hepatoprotective efficacy from the fruit extracts of *Diospyros malabarica* and *Gynocardia odorata* and understanding the possible mechanistic role in hepatoprotection.

## Materials and methods

### Chemicals

Methanol, DPPH, Ascorbic acid, Gallic acid, Hydrogen peroxide, and $CCl_4$ were procured from HiMedia Pvt. Ltd., India. RNA isolation kit and cDNA synthesis kit were procured from Life Technologies, India.

## Collection of plant material and extract preparation

The fruits of *D. malabarica* and *G. odorata* were procured from the province of Nilachal hill, Kamrup, Assam, India (26˚11′0″N 91˚44′0″E). Fruits were rinsed thoroughly with distilled water, carved, dried, powdered and stored at 4˚C for further analysis. The collected plants were subjected to harbarium preparation and followed by identification at GUBH, the Dept. of Botany, Gauhati University, Assam, India. Nevertheless, accession numbers were assigned to the individual plants as *D. malabarica* (Acc. No. 18071 dt.04.11.2015) and *G. odorata* (Ac.No. 18072dt.04.11.2015).

## Qualitative and quantitative estimation of phytochemicals

Methanolic extract (70% v/v) of fruit samples of *D. malabarica* and *G. odorata* were prepared to estimate the phytochemicals such as flavonoids, tannins, saponins and alkaloids qualitatively as described earlier by Trease and Evans [19]. Bio-macromolecules viz. total protein, total carbohydrates present in the fruit extracts were estimated as per Lowry's method and Anthrone method, respectively [20, 21]. The presence of ascorbic acid in the fruit extract was measured in 4% oxalic acid by titrating against the 2,6-dichlorophenolindophenoldye until pink color appeared for confirmation. Folin-Ciocalteau's approach was used to determine the total phenol contents in the fruit extract [22]. *In-vitro* antioxidant activities of quercetin and histopathological studies of liver were performed, and details incorporated in the S1 File.

## HPLC analysis of plant extracts

Quercetin (QR), a polyphenolic flavonoid compound, is found in large amounts in fruit extracts derived from plants, which protects against oxidative stress and hepatotoxicity [23]. The abundance of quercetin as an active component in the fruit extracts of *D. malabarica* and *G. odorata*, was determined using HPLC (SYS-LC-138, Systronics, India). A mobile phase was preparedusing a mixture of 0.1% (v/v) methanol and (65:35%, v/v) ortho-phosphoric acid. The adjusted flow rate of mobile phase was 1.0 ml/min in the column (4.6 mm × 250 mm × 5μM, HiQ Sil C18-HS) and temperature maintained at 28˚C, while the injection volume was kept at 20 μL. An isocratic elution was carried out per sample for 15 min.

## Animals

A total of 30 Wistar albino rats (either sex) (8 to 12 weeks old), weighing from 100 g to 120 g were used in this study. The rats were kept in animal house of Gauhati University and acute oral toxicity OECD/OCED (423) guideline for the testing of selected plant samples (OECD/OCED (423) was followed in cages (6 Wistar albino rats per group/3 male and 3 female) under normal laboratory condition of humidity, temperature (22–25 $^oC$) and light (12/12 hour light dark cycle.) [24]. Animal models were fed with Normal pellet diet.

**Ethics statement.**   All procedures performed in studies involving animal models were in accordance with the ethical standards of the institutional ethics committee (Animal ethical committee, Gauhati University, Ref. No. IAEC/PER/2014-2015/01; 08/05/2015).

## Experimental design

Acute toxicity study for purified flavonoid quercetin from the fruits extracts of *Diospyros malabarica* and *Gynocardia odorata* was carried out as per the OECD guidelines for testing chemicals [24]. The Wistar rats received doses of methanolic extracts of the fruits orally which was prepared one day prior to oral dose in various concentrations viz: 50, 100, 200, 500, 1000, 1500 and 2000 mg/kg body weight of the animals respectively. Animal lay-up conditions (described

**Table 1.  Experimental design in five different groups of rats including control, silymarin and plant extracts of *D. malabarica* and *G. odorata*.**

| Name of Groups | No. of rats in each group | Details of group | Treatment details |
|---|---|---|---|
| GROUP A | 6 | Negative control | Normal saline water |
| GROUP B | 6 | Negative control | CCl$_4$ |
| GROUP C | 6 | Positive control | CCl$_4$ + Silymarin (200 mg/kg body weight) |
| GROUP D | 6 | Treatment group | CCl$_4$ + quercetin *D. malabarica* (200 mg/kg body weight) |
| GROUP E | 6 | Treatment group | CCl$_4$ + quercetin *G. odorata* (200 mg/kg body weight) |

in the previous section) were maintained at least for 7 days prior to dosing with free access to water and food *ad libitum* for acclimatization to the laboratory conditions. Five groups (A-E) of 6 rats each of either sex were used for the CCl$_4$-induced hepatotoxicity model. CCl$_4$ (10% in liquid paraffin, 1 mL/kg per day for seven consecutive days) was administered orally to induce liver injury in the four groups (B-E) [4]. 200 mg/kg body weight each of *D. malabarica*, *G. odorata*, and Silymarin were administered orally to these groups of animals at 2, 24 and 48 h interval after the administration of the last dose of CCl$_4$ (Table 1).Two hours after the final dose of extracts, Silymarin and saline water, all the animals were sacrificed.

The resulting body weight after fasting of each animal was determined and the dose is calculated according to the body weight. Food was withdrawn for next 3–4 hours in rats after the extract has been administered. The animals were observed for toxic symptoms continuously for the first 4 hours after dosing followed by their mortality and behavioral response for 48 hours. This observation was followed daily for a total of 14 days. Individual weights of animals were determined before the fruit extracts were administered. Change in body weight was calculated and recorded each day. From the collected blood samples, serum was tested for liver markers such as total protein (TP), albumin (Alb), globulin (Glob), total bilirubin, SGOT, SGPT, etc. *Invitro* antioxidant activity of fruit extracts and the histopathological studies were performed and details have been given in S1 File.

## Gene expression profiling and real time PCR analysis

Expression profiling of gene markers i.e., hepatocyte growth factor (HGF), metalloproteinase tissue inhibitor (TIMP1) and Matrix metalloproteinase (MMP2) were carried out from the extracted liver of different groups of experimental rats. RNA was extracted using QIAamp RNA Blood Mini Kit from the frozen liver as suggested by the manufacturer. RNA quantification for purity was analyzed using a spectrophotometer (Cary50, Agilent, Germany) and followed by cDNA synthesis following similar methods employed elsewhere [17]. Two endogenous housekeeping genes, hypoxanthine peptidylprolyl isomerase A (Ppia) and phosphoribosyltransferase 1 (Hprt1) were used to compare the relative amount of the transcripts in all groups [25].

## Molecular redocking for validation of docking score

Liver cytochrome P450 2E1 (Cyp2E1) belongs to the family of cytochrome P450 enzyme that plays a vital role in toxin, alcohol, drug, lipid, and carcinogen metabolism [26]. Molecular docking studies were carried out between the target protein Cyp2E1, silymarin and quercetin. The three-dimensional coordinates of Cyp2E1 X-ray 1.8 Å resolution in pdb format was

downloaded from RCSB PDB repository (PDB ID: 3T3Z). Two other major cell signaling proteins JAK2 and BCL-2 were also investigated for understanding the mechanism of quercetin action. The coordinate files for JAK2 (PDB ID: 2B7A) with atomic resolution 2.00 Å and for BCL-2 (PDB ID: 4IEH) atomic resolution 2.10 Å were fetched from protein databank for further analysis. 3D structures of quercetin and silymarin were procured from public database pubchem (*https://pubchem.ncbi.nlm.nih.gov/compound/quercetin#section=Top*) in SDF format and subsequently converted in pdb format using OpenBabel 2.2.3 [27]. Autodock version v 4.2.1 was used for the molecular docking studies. During molecular docking studies, three replicates were performed having the total number of solutions computed 50 in each case, with population size 500, number of evaluations 2500000, maximum number of generations 27000 and rest the default parameters were allowed. After docking, the RMSD clustering maps were obtained by re-clustering with a clustering tolerance 0.5 Å,1 Å and 2 Å, respectively, in order to obtain the best cluster having lowest energy score with high number of populations.

In order to obtain accurate binding affinities for quercetin QM-Polarized Ligand Docking (QPLD) was performed using Schrodinger 2018–4 package (*https://www.schrodinger.com/qm-polarized-ligand-docking*). Quantum mechanics ligand docking gave accurate treatment of electrostatic charges to quercetin to avoid charge polarization induced by the Cyp2E1, BCL-2 and JAK2 environment. QPLD combines the docking power of Glide with the accuracy of QSite in QM/MM software. To perform QM docking, glide docking was executed within a grid size of (nx, ny, nz) Å = (200, 208, 200) for Cyp2E1, Å = (69, 80, 65) for BCL-2 and Å = (81, 98, 76) for JAK2 proteins followed by addition of QM charges using Jaguar tool embedded in the Schrodinger 2018–4 package. Then redocking was performed in high precision with an approximate ligand van der Waals spacing 0.8 Å and maximum atomic displacement 1.3 Å. A maximum of 10 ligand docking poses were generated with RMS deviation 0.5 Å.

## Molecular dynamics simulation (MD) and free energy landscape analysis

The MD simulations studies were carried out in triplicate on the QPLD dock complexes for BCL-2, JAK2 and Cyp2E1 with quercetin using the Desmond 2020.1 from Schrödinger, LLC. The triplicate samplings were made using same parameters for each MD run in order to obtain reproducibility of the results. The OPLS-2005 force field [28–30] and explicit solvent model with the SPC water molecules were used in this system [31]. Na+ ions were added to neutralize the charge. 0.15 M, NaCl solution was added to the system to simulate the physiological environment. Initially, the system was equilibrated using NVT ensemble for 100 ps to retrainover the protein-quercetin complex. Following this step, a short run of equilibration and minimization were carried out using NPT ensemble for 12 ps.The NPTensemble was set up using the Nose-Hoover chain coupling scheme [32] with temperature at 27˚C, the relaxation time of 1.0 ps and pressure 1 bar maintained throughout the simulations. A time step of 2 fs was used. The Martyna-Tuckerman–Klein chain coupling scheme [33] barostat method was used for pressure control with a relaxation time of 2 ps. The particle mesh Ewald method [34] was used for calculating long-range electrostatic interactions, and the radius for the coulomb interactions were fixed at 9Å. RESPA integrator was used for a time step of 2 fs for each trajectory to calculate the bonded forces. The root means square deviation (RMSD), radius of gyration (Rg), root mean square fluctuation (RMSF) and number of hydrogen (H-bonds) were calculated to monitor the stability of the MD simulations. Free energy landscape of protein folding on quercetin bound complex was measured using geo_measures v 0.8 [35]. Geo_measures include powerful library of g_sham and form the MD trajectory against RMSD and radius of gyration (Rg) energy profile of folding recorded in a 3D plot using matplotlib python package.

## Molecular Mechanics Generalized Born and Surface Area (MMGBSA) calculations

The binding free energy (ΔGbind) of the docked complexes during MD simulations of BCL-2, JAK2 and Cyp2E1 complexed with quercetin were estimated using the molecular mechanics generalized born surface area (MMGBSA) module (Schrodinger suite, LLC, New York, NY, 2017–4). The OPLS 2005 force field, VSGB solvent model, and rotamer search algorithms were used to define the binding free energy during the calculation [36]. The MD trajectories frameswere selected at each 10 ns interval after MD run. The Eq 1 was used to calculate the total free energy binding:

$$\Delta Gbind = Gcomplex - (Gprotein + Gligand) \qquad (1)$$

Where, ΔGbind = binding free energy, Gcomplex = free energy of the complex, Gprotein = free energy of the target protein, and Gligand = free energy of the ligand. The MMGBSA outcome trajectories were analyzed further for post dynamics structure modifications.

## Dynamic cross correlation and principal component (PCA) analysis

In order to analyze the domain correlations, dynamic cross correlation matrix (DCCM)was generated across all Cα-atoms for all the complexes during the MD simulation of 100 ns. PCA analysis was performed to extract the global motions of the trajectories during 100 ns simulation of BCL-2, JAK2 and Cyp2E1 complexed with quercetin. A covariance matrix was generated to calculate the PCA as described elsewhere [37]. 20 different conformational modes of principal component as the motion of trajectories were calculated and a comparison of first highest mode (PC1), with PC2, PC3 and PC20 analyzed for conformational analysis of the quercetin bound complex. Free energy landscape of protein folding on quercetin bound complex was measured using geo_measures v 0.8 [35]. Geo_measures include powerful library of g_sham and form the MD trajectory PC1, PC2, PC3 and PC20 mode were recorded in a 3D plot using matplotlib python package.

# Results

## Screening of phytochemicals in the fruit extracts of *D. malabarica* and *G. odorata*

The active phytochemicals screening in qualitative analysis revealed that the methanolic fruit extracts of *D. malabarica* and *G. odorata* were well endowedwith flavonoids, tannins, saponins, ascorbic acid and alkaloids. The other biochemicals were determined quantitatively: carbohydrate (%, w/v) 8.56 ± 0.20 and 9.25 ± 0.38, protein (%, w/v) 4.77 ± 0.17 and 3.52 ± 0.24, significant quantities of phenolic content (μg GAE/mg) 223.5 ± 0.26 and 206.14 ± 0.52 and ascorbic acid content (mg/100g) 55.57 ± 0.75 and 42.66 ± 0.83.

## HPLC analysis of plant extracts

HPLC analysis of *Diospyros malabarica* and *Gynocordia odorata* fruit extracts displayed the abundance of a peak at retention time 1.512 minute (Fig 1A and 1C) which corroborated with the standard quercetin exhibiting retention time at 1.513 minutes (Fig 1B). The purified fraction of quercetin was collected using a fraction collector (FRC-10A, Shimadzu, Japan) and the fractions were pulled into a final concentration of 25μg/mL. Purity of the extracted quercetin from fruit extracts were determined by comparing with the standard Quercetin in HPLC.

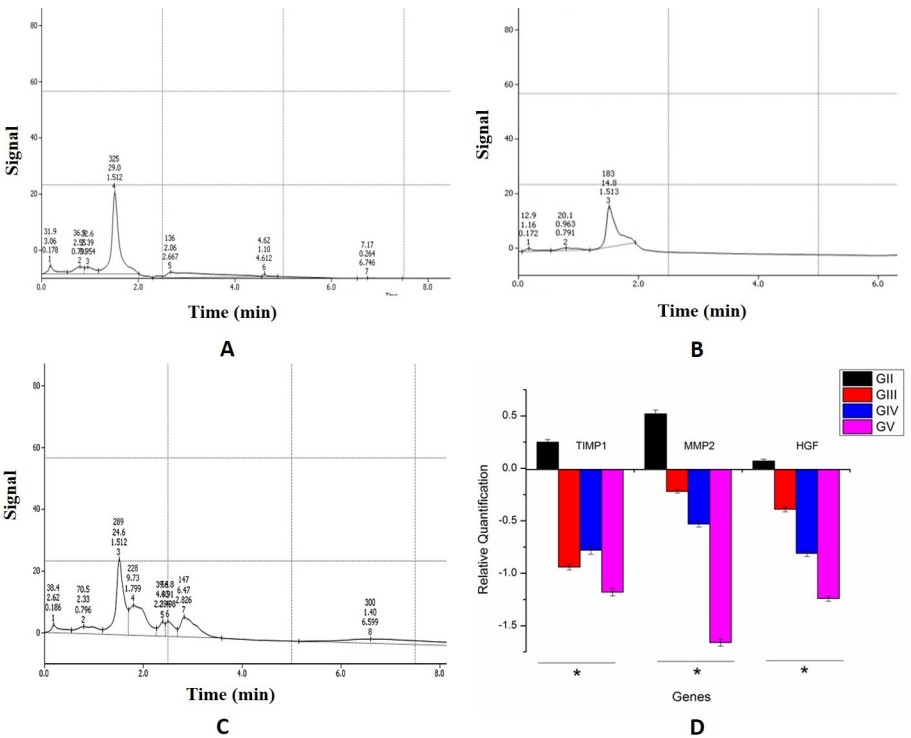

**Fig 1.** HPLC profile with retention time of quercetin compound from fruit extracts of (A) *Diospyros malabarica* and (C) *Gynocordia odorata* fruit extracts. (B) Profile of standard quercetin molecule. (D) Gene expression profile in real time PCR of quercetin on Group D and E rats.

Hence, quercetin is one of the abundant bioactive molecules present in the *D. malabarica* and *G. odorata* fruit extracts which might possesses the hepatoprotective activity.

## Liver function analysis

Experimentally designed groups of Wistar rats were observed continuously for any abnormalities as a result of toxicity such as writhing, gasping, palpitation and respiratory rate, or mortality. The liver function analysis is displayed in Table 2. Histopathology study was explained in detail in S1 File. The oral dose of *Diospyros malabarica* and *Gynocordia odorata* fruit extracts did not display any mortality neither showed any signs of toxicity at an applied dose 2000 mg/kg body weight. The effect of fruit extracts from *Diospyros malabarica* and *Gynocordia odorata* displayed lowering of bilirubin 0.41±0.007 g/dl and 0.45±0.005, SGOT 127.25±2.02 and 132.15 ±2.17 U/mL, SGPT 83.58±1.78 and 87.82±1.56 U/mL, respectively, as compared to the

**Table 2. Effects of methanolic extract from *Diospyros malabarica* and *Gynocardia odorata* on liver Total protein, Albumin, Globulin, Total bilirubin, SGOT and SGPT on CCl₄ induced hepatotoxicity in rats.**

| Groups | Total protein (g/dL) | Albumin (g/dL) | Globulin (g/dL) | Total bilirubin (g/dL) | SGOT (U/mL) | SGPT(U/mL) |
|---|---|---|---|---|---|---|
| Control | 6.84±0.14 | 3.95±0.37 | 3.28±0.14 | 0.26±0.014 | 80.64±1.22 | 41.98 ±1.60 |
| CCl₄ treated (1mL/kg) | 5.36±0.12 | 2.62±0.22 | 2.60±0.06 | 0.69±0.009 | 189.20±3.95 | 102.42±1.59 |
| CCl₄ + Silymarin (200 mg/kg) | 6.47±0.06 | 3.28±0.10 | 3.19±0.06 | 0.32±0.004 | 109.12±1.51 | 69.54±2.01 |
| CCl₄+ quercetin *D. malabarica* (200 mg/kg) | 6.16±0.26 | 3.12±0.06 | 3.00±0.26 | 0.41±0.007 | 127.25±2.02 | 83.58±1.78 |
| CCl₄+ quercetin *G. odorata* (200 mg/kg) | 5.94±0.31 | 2.98±0.08 | 3.13±0.21 | 0.45±0.005 | 132.15±2.17 | 87.82±1.56 |

hepatotoxic rats treated with CCl₄ and the results were quite comparable with silymarin treated rats. In some cases, e.g., the quantities of SGPT and SGOT, in fact,were comparatively lower than the commercial drug silymarin treated rats. On the other hand, quantities of total protein, albumin and globulin did not show much significant variance among the control and treated groups (Table 2). Significant free radical scavenging activities of quercetin present in *Diospyros malabarica* and *Gynocordia odorata* fruit extracts against DPPH and $H_2O_2$ were also observed (S1 File). Histopathological studies of rat liver displayed the ameliorating pattern after treatment with quercetin containing *Diospyros malabarica* and *Gynocordia odorata* fruit extracts (S1 File).

## Gene expression profiling

**Integrity of RNA.** The 260/280 ratio for the RNA isolated from the liver tissue samples ranged from 2.08–2.14 suggesting good quality RNA (Table ST3 in S2 File). The integrity of RNA was checked on agarose gel showing discrete 28S and 18S ribosomal RNA band on each sample suggesting that the RNA in each case was intact and could be used for qPCR analysis (Fig S3 in S2 File).

## Real time PCR analysis

In the qPCR analysis, high expression of genes, hepatocyte growth factor (HGF), the tissue inhibitors metalloproteinase (TIMP1) and Matrix metalloproteinase (MMP2) from the liver tissue of animal models were observed in the CCl₄ treated animals (Group B) (Fig 1D). In contrast, down regulation of these were noted for positive control (Group C), treated with commercial drug silymarin as well as in the Group D treated with quercetin obtained from *D. malabarica* and *G. odorata* (Group E) (Fig 1D). Hprt1 and Ppia were used as housekeeping genes for this study. The qPCR outcome was examined using the students t-test (P< = 0.05) (Group B versus other groups individually).

## Molecular redocking for validation of docking score

In molecular docking analysis of BCL-2, JAK2 and Cytochrome P450 Cyp2E1 with the ligand quercetin in Autodock output, the best conformation was displayed in a dock complex (Table 3). The best dock pose was seleccted based on low RMSD tolerance 0.5 Å and binding energy having maximum within that RMSD cluster. BCL-2-quercetin complex showed free energy of binding (Δ;G) -8.7 kcal/mol, inhibitory concentration 5.06 μM, ligand efficiency -0.5, total internal energy -3.1 kJ/mol, and tortional energy 1.25 kJ/mol. The principal residues making the binding pocket around quercetin is comprised of Phe97, Tyr101, Phe105, Leu108, Glu129, Leu130, Asp133, Arg139, Ala142, Ser145 (Fig 2A). On the other hand, JAK2-quercetin complex displayed free energy of binding (Δ;G) -8.8 kcal/mol, inhibitory concentration 1.14 μM, ligand efficiency -0.11, total internal energy -4.1 kJ/mol, and tortional energy 0.72 kJ/

**Table 3. Binding energy properties of BCL-2, JAK2 and Cyp2E1 with quercetin in docking and redocking in QPLD.**

| Quercetin docked with | $\Delta G_{bind}$ (kcal/mol) | $\Delta G_{bind}$(QPLD) (kcal/mol) | Ki (μM) | Residues at the binding site cavity |
|---|---|---|---|---|
| BCL | -8.7 | -11.41 | 5.06 | Phe97, Tyr101, Phe105, Leu108, Glu129, Leu130, Asp133, Arg139, Ala142, Ser145 |
| JAK | -8.8 | -12.02 | 1.14 | Leu828, Gly829, Phe833, Lys855, Glu871, Leu900, Met902, Tyr904, Leu905, Gly908, Cys909, Arg953, Asn954, Asp967 |
| Cyp2E1 | -6.05 | -8.71 | 22.01 | Leu313, Met316, Pro462, Leu463, Val464, Asp468, ILE469, Asp470, Pro483, Tyr485, Lys486 |

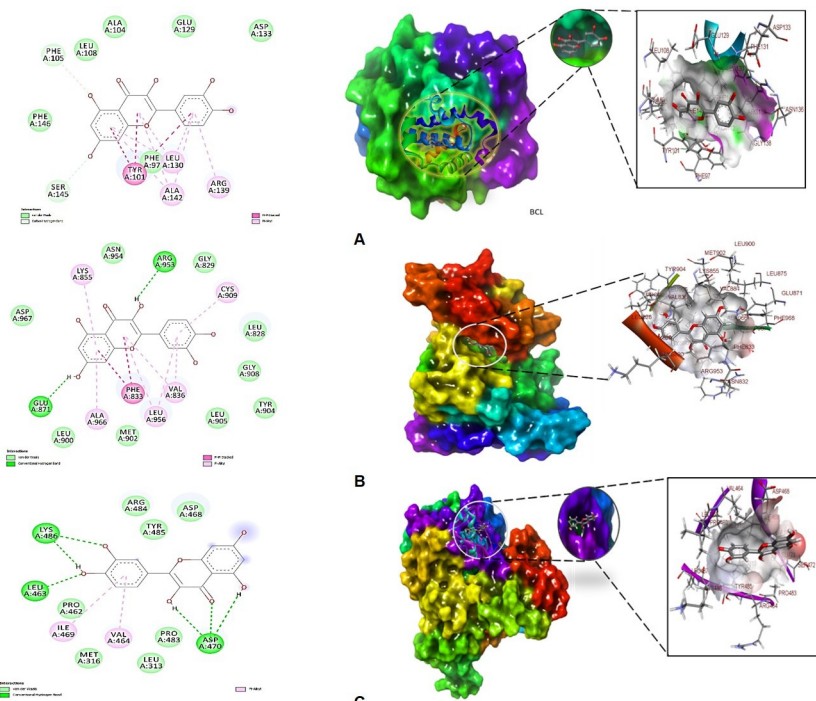

**Fig 2.** Best docked pose of quercetin with (A) BCL-2, (B) JAK2 and (C) Cytochrome P450 Cyp2E1 displaying 2D interaction plot on the left panel. Green dashed lines indicating the conventional hydrogen bonds and residues embedded in light green sphere indicating to involve in hydrophobic interactions. On the center panel, surface view of (A) BCL-2, (B) JAK2 and (C) Cytochrome P450 Cyp2E1 displaying binding cavity of quercetin and right panel displaying the zoomed out binding pocket having amino acid residues at 3Å surrounding the quercetin molecule.

mol. Residues conforming the quercetin binding pocket are Leu828, Gly829, Phe833, Lys855, Glu871, Leu900, Met902, Tyr904, Leu905, Gly908, Cys909, Arg953, Asn954, Asp967, however, Glu871 and Arg953 involved in forming conventional hydrogen bonds (Fig 2B).

The quercetin bound to Cyp2E1 with convincing binding energy (Δ;G) -6.05 kcal/mol, inhibitory concentration (Ki) 22.01 μM, ligand efficiency -0.29, total internal energy -1.8 kJ/mol and tortional energy 2.09 KJ/mol. Quercetin (ligand) in the Cyp2E1 complex exhibited Leu313, Met316, Pro462, Leu463, Val464, Asp468, ILE469, Asp470, Pro483, Tyr485, Lys486 are major amino acid residues involved in the formation of binding cavity (Fig 2C). However, Leu463, Asp470 and Lys486 were involved in conventional hydrogen bonds (Fig 2C, right panel 2D plot). The quercetin-BCL-2, JAK2 and Cyp2E1 complex docking energies were recalculated using glide, QPLD with QM and MM optimized redocking following extra precision protocol (XP) showed -11.41, -12.0 and -8.71 kcal/mol binding free energies at the same binding site used in Autodock tool, respectively. Therefore, validated docking scores confirmed significant binding of quercetin with diverse liver targets involved in cell signaling. Binding energies suggested that quercetin has good affinity for the target proteins BCL-2, JAK2 and Cyp2E1.

## Molecular dynamics simulation (MD) and free energy landscape analysis

Molecular dynamics and simulation (MD) studies were carried out to determine the stability and convergence of quercetin bound protein complexes. Each simulation of 100 ns displayed stable conformation while comparing the root mean square deviation (RMSD) values. The Cα-backbone of BCL-2 bound to quercetin exhibited a deviation of 1.1 Å (Fig 3A), while JAK2

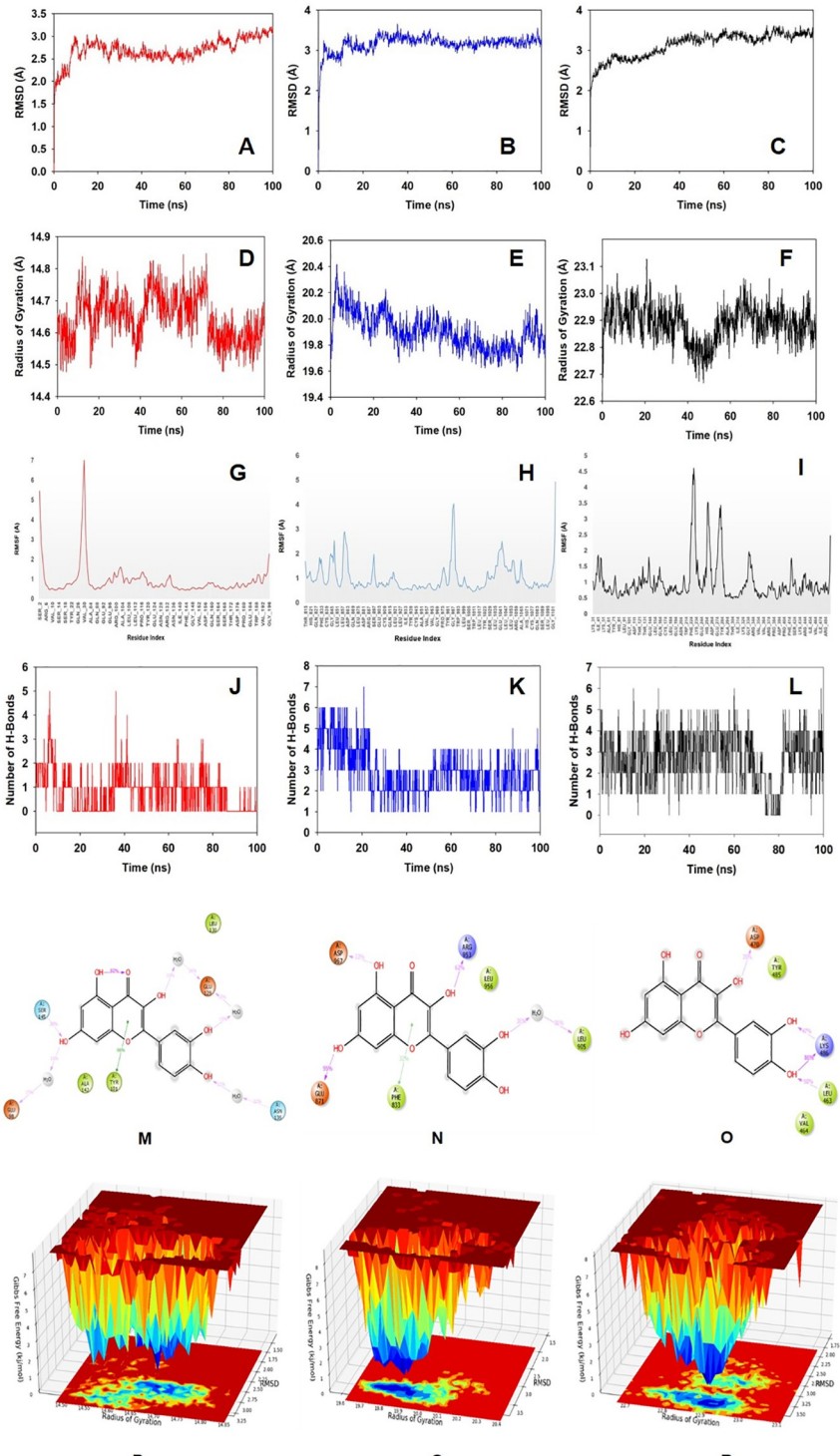

**Fig 3. Analysis of MD simulation trajectories for 100 ns.** RMSD plots displaying the molecular vibrations of Cα backbone of (A) BCL-2 (B) JAK2 and (C) Cyp2E1. Radius of gyration plots for the deduction of compactness of protein (D) BCL-2, (E) JAK2 and (F) Cyp2E1. RMSF plots showing the fluctuations of respective amino acids throughout the simulation time 100 ns for (G) BCL-2, (H) JAK2 and (I) Cyp2E1. Number of Hydrogen bonds formed during the course of simulation between quercetin and (J) BCL-2, (K) JAK2 and (L) Cyp2E1. 2D interaction plot of post simulation time between the quercetin and (M) BCL-2 (N) JAK2 and (O) Cyp2E1. Free Energy Landscape displaying the achievement of global minima (ΔG, kj/mol) of (P) BCl-2 (Q) JAK2 and (R) Cyp2E1 in presence of quercetin with respect to their RMSD (Å) and radius of gyration (Rg, Å).

displayed comparatively stable 0.5 Å deviation (Fig 3B). On the other hand, Cα-backbone of Cyp2E1 displayed more RMS deviation as compared to BCL-2 and JAK2 with 1.5 Å fluctuation (Fig 3C). RMSD plots are within the acceptable range signifying the stability of proteins in the quercetin bound state before and after simulation.It can also be suggested that quercetin bound BCL-2, JAK2 and Cyp2E1 are quite stable in complex due to higher affinity of the ligand. Radius of gyration is the measure of compactness of the protein. Here in this study, BCL-2 backbone displayed less fluctuating radius of gyration (Rg) initially till 80 ns of simulation while later upto 100 ns became stable (Fig 3D). JAK2 backbone displayed the lowering of Rg till 85 ns but later went up to regain the compactness of the protein (Fig 3E). On the other hand, stable Rg observed in case of Cyp2E1 except a deep angle at 45 ns which has regained its shape thereafter and thus confirming significant compactness of the protein in quercetin bound state (Fig 3F). The overall quality analysis from RMSD and Rg it can be suggested that quercetin bound to the protein targets posthumously in the binding cavities and played a significant role in stability of the proteins. The plots for root mean square fluctuations (RMSF) displayed significant spike of fluctuation (7 Å) of amino acid residues Gln26 and Val30 in BCL-2 protein while the rest of the residues less fluctuating during the entire100 ns simulation (Fig 3G). RMSF plot of JAK2 displayed less fluctuation of residues of 3–4 Å indicating the stable amino acid conformations during the simulation time (Fig 3H). Cyp2E1 displayed significant amino acid fluctuations from Trp214 to Tyr274 residues while the rest of the residues were less fluctuating (Fig 3I). Therefore, for RMSF plots it can be suggested that the proteins structures were stable during simulation in quercetin bound conformation. Quercetin formed single conventional hydrogen bond with Ser145 of BCL-2 protein at a frequency of 38% throughout simulation time. While water bridges, pi-pi and hydrophobic interactions also helped to form stable complex (Fig 3J). Quercetin bound to JAK2 with significant numbers of conventional hydrogen bonds as displayed in 2D interaction plot (Fig 3K). Glu871, Arg953 and Asp967 are the key residues in JAK2 protein formed hydrogen conventional hydrogen bonds at a frequency 95%, 62% and 33%, respectively with quercetin. On the other hand, quercetin formed hydrogen bonds with Leu463 with 50%, Asp470 with 35% and couple of hydrogen bonds with Lys486 via–OH groups of quercetin with frequencies 86% and 47% throughout the 100 ns simulation time (Fig 3L).The non-bonded interactions played critical role in quercetin and protein complex integrity. Quercetin formed an average of single hydrogen bond with BCL-2 (Fig 3M) whereas with JAK2 confined to average 3 hydrogen bonds (Fig 3N). While Cyp2E1 displayed conventional hydrogen bond with Asp470 and Lys486 (Fig 3O). The 2D quercetin and binding cavity residues interaction plots displayed good agreements with the outcome of hydrogen bonds formation.

The free energy landscape of (FEL) of achieving global minima of Cα backbone atoms of proteins with respect to RMSD and radius of gyration (Rg) are displayed in Fig 3. BCL-2 bound to quercetin achieved the global minima (lowest free energy state) at 2.75 Å and Rg 14.65 Å (Fig 3P). The FEL envisaged for deterministic behaviour of BCL-2 to lowest energy state owing to its high stability and best conformation at quercetin bound state. While one the other hand, JAK2 Cα backbone atoms conformed into lowest energy state at RMSD 3.4 Å and Rg 19.8 Å to achieve global minima and stable state (Fig 3Q). Similarly, Cyp2E1 exhibited global minima state at 3.5 Å andRg 22.9 Å (Fig 3R). FEL is the indicator of the protein folding to attain minimum energy state and that aptly achieved due to quercetin bound state.

The energy profiles of the protein quercetin complex systems were determined to display the stability of the entire system. In this regard, the total energy (ETOT) of the BCL-2 quercetin system shownto be very stable with an average total energy -45 kcal/mol (Fig 4A, red). However, van der Waal's energy (vdW) displayed to be merged over the total energy with an average energy -35 kcal/mol, contemplating as principal contributor to the stability of the

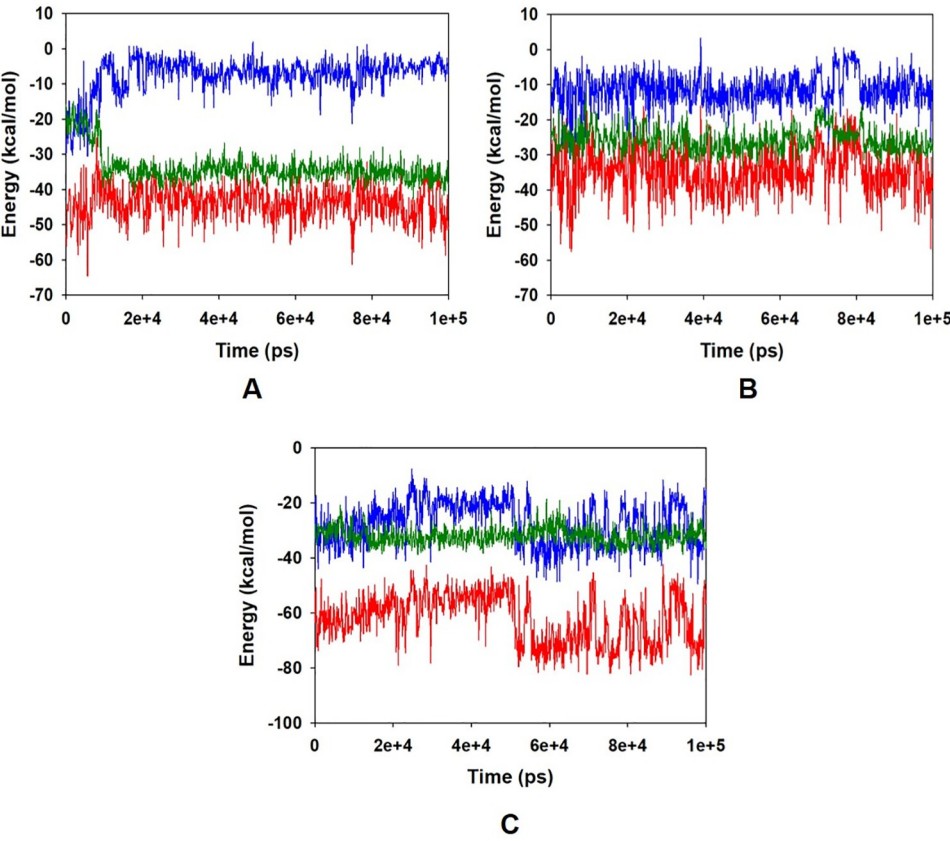

**Fig 4.** Energy plot of protein (A) BCL-2 (B) JAK2 and (C) Cyp2E1 and quercetin complex system during the entire simulation event of 100 ns. The total energy (red), van der Waal's energy (green) and coulomb energy (blue) of the entire system indicating the stability of the individual systems bound to quercetin molecule.

BCL-2 quercetin complex (Fig 4A, green). In addition, coulombic interactions played minor role in the system stability and contributing an average energy -5.0 kcal/mol (Fig 4A, blue). Energy profile of JAK2-quercetin complex displayed an average of -40 kcal/mol of ETOT (Fig 4B, red), while vdW energy contributed -25 kcal/mol (Fig 4B, green) and coulombic interaction -10 kcal/mol (Fig 4B, blue). Similar behavior was also observed in case of Cyp2E1 bound quercetin system where average ETOT was measured to be -65 kcal/mol (Fig 4C, red), and contributing vdW (Fig 4C, green) and coulombic energies (Fig 4C, blue) were -22 kcal/mol and -30 kcal/mol, respectively. The high negative values indicating lowest potential energy in the individual systems to achieve global minima of protein-quercetin complex.

## Molecular Mechanics Generalized Born and Surface Area (MMGBSA) calculations

MMGBSA is a popular method in calculating the binding energy of ligand to protein molecules. The estimation of the binding free energy of each of the protein-quercetin complexes, as well as the role of other non-bonded interactions energies were estimated. It is evidenced from Table 4, the binding free energy ($\Delta$Gbind) of proteins BCL-2, JAK2 and Cyp2E1 and quercetin complex, The average binding energy of the ligand quercetin with BCL-2-41.48±11.02 kcal/mol, while with JAK2–43.53±6.55 kcal/mol and with Cyp2E1–39.89±5.78 kcal/mol. The $\Delta$Gbind is influenced by of various types of non-bonded interactions, including $\Delta$Gbind

**Table 4. Binding energy calculation of quercetin with BCL-2, JAK2 and Cyp2E1 and non-bonded interaction energies from MMGBSA trajectories.**

| Energies (kcal/mol)* | BCL-2 | JAK2 | Cyp2E1 |
|---|---|---|---|
| $\Delta G_{bind}$ | -41.48±11.02 | -43.53±6.55 | -39.89±5.78 |
| $\Delta G_{bind}Lipo$ | -9.20±1.02 | -9.98±0.76 | -9.90±0.75 |
| $\Delta G_{bind}vdW$ | -35.97±5.77 | -37.49±3.34 | -26.91±1.28 |
| $\Delta G_{bind}Coulomb$ | -8.48±3.64 | -16.97±4.73 | -12.32±4.24 |
| $\Delta G_{bind}H_{bond}$ | -0.92±0.65 | -2.15±0.42 | -2.10±0.55 |
| $\Delta G_{bind}SolvGB$ | 15.48±5.63 | 23.14±2.37 | 8.10±1.52 |
| $\Delta G_{bind}Covalent$ | 0.73±0.20 | 1.88±0.47 | 3.28±0.87 |

*Results are calculated in mean ± SD.

Coulomb, ΔGbindCovalent, ΔGbindHbond, ΔGbindLipo, ΔGbindSolvGB and ΔGbindvdW interactions. Among all the types of interactions ΔGbindvdW, ΔGbindLipo and ΔGbindCoulomb energies contributed most to achieve the average binding energy. In contrast, ΔGbind-SolvGB and ΔGbindCovalent energies contributed the lowest to attain the final average binding energies. In addition, the values of ΔGbindHbond interaction of quercetin protein complexes showed the stable hydrogen bonds with the amino acid residues. In all the complexes ΔGbindSolvGB and ΔGbindCovalent showed unfavorable energy contributions and thus opposed binding. It is observed from Fig 5A (left panel), at pre-simulation (0 ns) quercetin at the binding pocket of BCL-2 undergone substantial angular movement of the pose (curved to straight) after post simulation (100 ns) (Fig 5A, right panel). However, in JAK2 binding cavity, the ligand quercetin during pre-simulation was quite flat (Fig 4B, left panel) but later after post-simulation (at 100 ns) observed to be change in the tortional angles with an opened conformation facing toward the pocket (Fig 5B, right panel). On the other hand, Cyp2E1 bound quercetin displayed relative movement form initial position 0 ns to final 100 ns trajectory (Fig 5C, right and left panel). These conformational changes consequences the better acquisition at the binding pocket as well as the interaction with the residues for higher stability and better binding energy.

Thus MMGBSA calculations resulted, from MD simulation trajectories well corroborated with the binding energies calculated from the docking results. Therefore, it can be suggested that the quercetin molecule has good affinity for the major targets BCL-2, JAK2 and Cyp2E1. The MMGBSA trajectories displayed the conformational changes in the quercetin to achieve the best fitting in the binding cavity of the protein.

## Dynamic cross correlation and principal component (PCA) analysis

MD simulation trajectories are analysed for dynamic cross correlation among the domains within protein chains bound with quercetin molecule. For correlative dynamic motion, the cross correlation matrices of BCL-2, JAK2 and Cyp2E1 were generated and displayed in Fig 6. The blue blocks displayed in the plot indicated the residues having high correlated movement and red having least correlation. The amino acid residues of quercetin bound BCL-2 showed concerted movement of residues (Fig 6A) coformed into a α-helix (5–25, red), residues 82–122 (green) conformed into two α-helices and 125–135 conformed into partial α-helix and loop (magenta) (Fig 6A). On the other hand, quercetin bound JAK2 diplayed 92–150 residues highly correlated movement and conformed into a loop (red) (Fig 6B). However, quercetin bound Cyp2E1 displayed small blocks of correlated motion from residues 1–50, 80–100, 382–

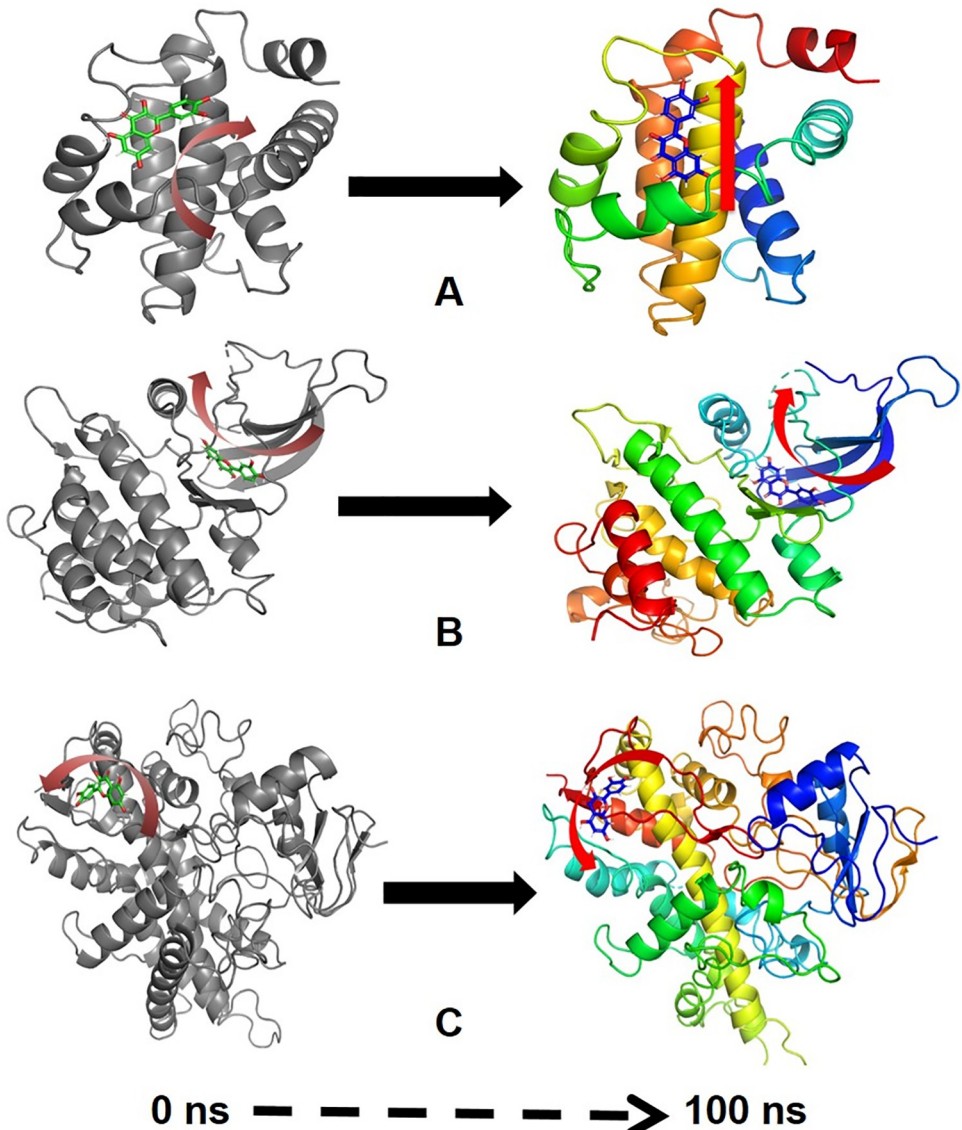

**Fig 5.** MMGBSA trajectory (0 ns, before simulation and 100 ns, after simulation) exhibited conformational changes of quercetin upon bidning with the proteins (A) BCL-2, (B) JAK2 and (C) Cyp2E1. The arrows indicating the overall positional variation (movement and pose) of quercetin at the bidning site cavity.

385 (Fig 6C) conformed into loop (red), α-helix (green) and a helical turn (magenta), respectively (Fig 6C).

Principal component analysis (PCA) of the MD simulation trajectories for BCL-2, JAK2 and Cyp2E1 bound to quercetin molecule was analyzed to interpret the randomized global motion of the atoms of amino acid residues. This analysis interprets the more flexible scattered trajectories owing the distortion of the protein structure. The internal coordinates mobility into three-dimensional space in the spatial time of 100 ns were recorded in a covariance matrix and rational motion of each trajectories are interpreted in the form of orthogonal sets or Eigen vectors. PCA analysis of BCL-2.

MD simulation trajectory Cα atoms displayed scattered unordered orientation owing to their less equilibrated form in first three modes. The first highest mode (PC1) displayed 28.7%

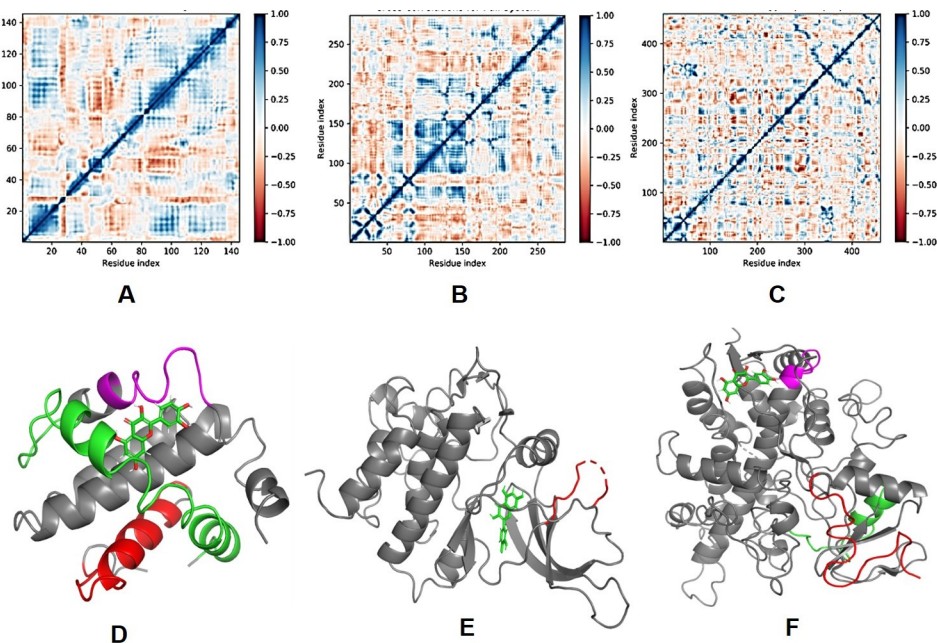

**Fig 6.** Dynamic Cross Correlation matrix (DCCM) of (A) BCL-2, (B) JAK2 and (C) Cyp2E1 and correlated amino acids conformed into secondary structural domains (colored) and non-correlated domains (grey) of (D) BCL-2, (E) JAK2 and (F) Cyp2E1 proteins bound with quercetin (green).

of the trajectories having 64.024 variance with least coordinated aggregate motion 28.717. While in the second mode (PC2) high variance 34.762 among the 15.9% trajectories with an aggregated motion 43.12 and the third mode (PC3) variance 20.106 is found among 9.0% trajectories with aggregated motion 53 (Fig 7A). However, the PC20 described very less variance 1.312 among 0.59% trajectories with aggregated motion 84.638. The combined plots of all the three PC modes displayed in Fig 7A. It is also observed from the PCA plots, as the sampling size increased upto PC mode 20 the trajectories are more aligned (Fig 7A, green). Following BCL-2 complex, quercetin bound JAK2 Cα atoms trajectory analysis displayed three distinct scattered clusters in PC1, PC2 and PC3 modes with 23.43%, 13.93% and 9.42%, respectively (Fig 7B). The variance in each mode was calculated as 90.623, 53.894 and 36.456 and aggregated motion 23.43, 37.37 and 46.80, respectively for PC1, PC2 and PC3. However, at mode PC20 the scatteredness reduced to a uniform cluster to 0.703%, variance 2.732 and aggregated motion 78.08. On the other hand, quercetin bound Cyp2E1 displayed 31.21%, 12.97% and 5.18% contribution to scattered motion of trajectories in PC1, PC2 and PC3 modes. Average variance was observed to be 197.927, 82.291 and 32.898 and aggregated motions of the trajectories were recorded 31.21, 44.195 and 49.384 respectively at PC1, PC2 and PC3. While in PC20 (Fig 7C) the observed scattered 31.21% with variance 3.398 and aggregated motion 77.822. Therefore, PCA analysis suggested that the Eigen vectors of relative aggregated motion of the trajectories became better at higher mode PC20 into a converted global motion of the trajectories during simulation indicating high ordered protein structure and conformation during quercetin bound state. Moreover, it can be suggested that the complex between quercetin and BCL-2, JAK2 and Cyp2E1 are very stable complex. The FEL between the highest mode (PC1) and lowest (PC20) in protein conformational variation are displayed in Fig 7D.

The protein conformational stability of BCL-2 in quercetin bound state achieved due to the positive correlation motion of the MD trajectories. Moreover, highest and lowest PCA modes

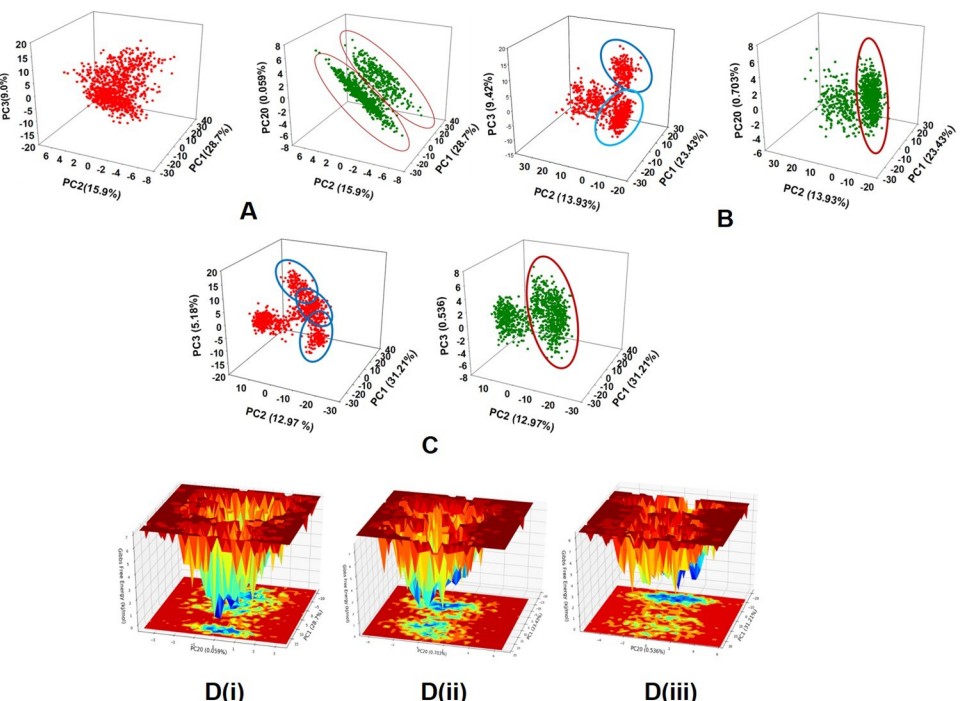

**Fig 7. PCA plots of four PC modes where PC1 the highest variance and PC20 the lowest variance Eigen values.**
PC2 and PC3 are the second and third highest variances respectively. The 3D PCA plots of among PC1, PC2 and PC3 (in left panel) and PC1, PC2 and PC20 (right panel) displayed for (A) BCL-2, (B) JAK2 and (C) Cyp2E1, bound to quercetin molecule. The round up zones in the PCA plots displaying the clustered trajectories in the respective modes. The representation of free energy landscape (FEL) against PC1 and PC2 for the proteins (D) (i) BCL-2, (ii) JAK2 and (iii) Cyp2E1, respectively.

exhibited the more feasible and quicker thermodynamically favorable conformation of BCL-2 [Fig 7D (i)]. Whereas the FEL for JAK2 against the dominant PCA modes exhibited quick and thermodynamically favorable protein folding [Fig 7D (ii)]. This behavior is observed due to correlated motions among the dynamic trajectories in JAK2 [Fig 7D (ii)]. Cyp2E1 showed interesting FEL where the PCA mode contribution attributed for quicker thermodynamically favorable protein folding and stabilizing the conformation [Fig 7D (iii)]. Therefore, it can be suggested from the free energy landscape of proteins in quercetin bound state achieve high favorable conformations meant for higher binding and stable complex.

## Discussion

Medicinal plants have phytochemicals which act as active principle against hepatio-toxins in prevention of hepatic injury. *Gynocardia odorata* R. Br. and *Diospyros malabarica* (Ders.) Kostel are two important plants endowed with many medicinal components. The present study encompasses the presence of active flavonoid compound "quercetin" in the methanolic fruit extracts from *G. odorata* and *D. malabarica*. Fruit extracts of *G. odorata* and *D. malabarica* exhibited comparable antioxidant activity as reported from the plant extracts elsewhere [16, 38]. Earlier investigations have revealed that the antioxidant activities of phytochemicals have significant impact on hepatoprotection [38–40]. Antioxidant scavenging activity against DPPH and $H_2O_2$ of active phytochemical in the fruit extracts *G. odorata* and *D. malabarica* were assessed in this current study (S1 File). The significant hepatoprotective activities of the

active flavonoid compound quercetin from the *G. odorata* and *D. malabarica* were observed against CCl$_4$ persuaded hepatic injury in rats by enumerating the total serum albumin, globulin, and bilirubin. These findings were quite comparable with the positive control silymarin. Maintenance of the total serum albumin, globulin, bilirubin levels in blood serum indicates the improved liver condition while comparing with a healthy liver. On the other hand, the lowered levels of SGOT and SGPT in the quercetin treated blood serum content as compared to CCl$_4$ intoxicated mice signifies the improved level of hepatic health. This finding is also corroborated with the SGPT and SGOT levels of silymarin treated rats. Similar study has been reported earlier where the aqueous extracts of *Curcuma longa* down regulate the serum SGPT and SGOT during hepatoprotection in CCl$_4$ induced hepatic injury [40]. Histopathological studies of quercetin-treated liver tissues exhibited the reduction of polynuclei, granular cytoplasm and regeneration of blood vessels as well as hepatocytes in the CCl$_4$ induced liver (S1 File). Similar observation was found in case of commercial drug silymarin treated liver. Analogous report also suggested that extract of *Homalium letestui* stem against paracetamol-induced liver injury improved the liver morphology by rearrangements of blood vessels and reduction of inflammatory cells [41, 42]. The upregulation of Hepatocyte growth factor (HGF), the tissue inhibitors metalloproteinase (TIMP1) and matrix metalloproteinase (MMP2) genes Haf, timp1 and mmp2, respectively, in the hepatocytes play a major role in liver fibrosis and damage [42, 43]. A contrasting dominance of down regulation of HGF and metalloproteinase in quercetin-treated liver injured with CCl$_4$ leading to hepatoprotection has confirmed the regeneration of hepatic cells. In silico studies by molecular docking, molecular dynamics and simulations suggested that the quercetin molecule binds with significant binding energy with hepatic microsomal Cytochrome P450 Cyp2E1 which is a major target site for drug metabolism and detoxification. During liver injury by the induction of CCl$_4$, the instability of Cytochrome p450 Cyp2E1 protein is leading to enormous oxidative stress and dysfunction [44]. While the in-silico analysis exhibited that quercetin molecule stabilizes the molecular architecture by making a compact orientation of the constitutive amino acids at the binding site. That leads to a stable conformation of the Cyp2E1 protein which may help to detoxification of CCl$_4$ rapidly and to regain the hepatic health. In addition, quercetin molecule showed significant stable binding with BCL-2 and JAK2 regulatory proteins of cell signaling proteins. BCL-2 activation may lead to deleterious effect on hepatocarcinoma leading to cell death [45]. Therefore, inhibition of BCL-2 leading to prevention of hepatocarcinoma in insilico MD simulation gave suggestive approach for invitro and invivo studies [45]. Similarly, quercetin, in this study exhibited significant binding, stability, and inhibition of BCL-2 in MD simulations corroborated the ameliorating effect in CCl$_4$ induced hapatocarcinoma in rats. Another route to hepatocarcinoma induction by chemical agents is due to JAK2 autophosphorylation leading to inability of STAT3 binding to DNA [46]. In addition, Zhong and coworkers reported the in silico prediction about JAK2 inhibition by natural products and ameliorating activity of hepatocarcinoma [46]. Similar findings have corroborated our results on quercetin displayed significant inhibition in molecular docking. MD simulation studies and free energy landscape (FEL) of JAK2 while bound to quercetin deciphered the good stability of the complex and plausibly allowed to predict quercetin as potent inhibitor against JAK2 during hepatoprotection.

The quercetin-induced stability of the Cyp2E1, BCL-2 and JAK2 perhaps unlocked a novel theragnostic approach for diminishing the effects of CCl$_4$ and reduced hepatotoxicity. Therefore, a detailed study on the dynamic functions of quercetin compound from *G. odorata* and *D. malabarica* may pave a potential route towards the identification of a new drug molecule for hepatoprotective function.

## Supporting information

**S1 File.**
(PDF)

**S2 File.**
(PDF)

## Acknowledgments

Authors are thankful to DST FIST support, Govt. of India to the Department of Botany, Gauhati University and also thankful to Department of Bioengineering and Technology, Gauhati University for proving conducive environment and support to carry out this work. This work is funded by Researchers Supporting Project (RSP- 2021/26), King Saud University, Riyadh, Saudi Arabia.

## Author Contributions

**Conceptualization:** Arabinda Ghosh, Pranjal Sarmah, Debabrat Baishya.

**Data curation:** Debabrat Baishya.

**Formal analysis:** Arabinda Ghosh, Pranjal Sarmah, Debabrat Baishya.

**Funding acquisition:** Rajbardhan Mishra, Saad Alkahtani.

**Investigation:** Pranjal Sarmah, Nobendu Mukerjee, Debabrat Baishya.

**Methodology:** Arabinda Ghosh, Pranjal Sarmah, Debabrat Baishya.

**Project administration:** Debabrat Baishya.

**Resources:** Harun Patel.

**Software:** Arabinda Ghosh.

**Supervision:** Debabrat Baishya.

**Validation:** Rajbardhan Mishra, Debabrat Baishya.

**Writing – original draft:** Arabinda Ghosh, Rajender S. Varma.

**Writing – review & editing:** Arabinda Ghosh, Rajbardhan Mishra, Saad Alkahtani, Rajender S. Varma, Debabrat Baishya.

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
