## [Decision Letter · Decision Letter 0]

22 Nov 2021

PONE-D-21-29456Nonlinear Molecular Dynamics of quercetin: Its mechanistic role in hepatoprotectionPLOS ONE

Dear Dr. Baishya,

Thank you for submitting your manuscript to PLOS ONE. After careful consideration, we feel that it has merit but does not fully meet PLOS ONE’s publication criteria as it currently stands. Therefore, we invite you to submit a revised version of the manuscript that addresses the points raised during the review process.

The authors are advised to address all the comments carefully.

We look forward to receiving your revised manuscript.

Kind regards,

Ghulam Md Ashraf, Ph.D.

Academic Editor

PLOS ONE

Journal Requirements:

"No"

 This information should be included in your cover letter; we will change the online submission form on your behalf

Reviewers' comments:

Reviewer's Responses to Questions

**Comments to the Author**

1. Is the manuscript technically sound, and do the data support the conclusions?

Reviewer #1: Yes

Reviewer #2: Yes

Reviewer #3: Yes

Reviewer #4: Partly

Reviewer #5: Yes

2. Has the statistical analysis been performed appropriately and rigorously? 

Reviewer #1: Yes

Reviewer #2: Yes

Reviewer #3: Yes

Reviewer #4: I Don't Know

Reviewer #5: Yes

3. Have the authors made all data underlying the findings in their manuscript fully available?

Reviewer #1: Yes

Reviewer #2: Yes

Reviewer #3: Yes

Reviewer #4: No

Reviewer #5: Yes

4. Is the manuscript presented in an intelligible fashion and written in standard English?

Reviewer #1: Yes

Reviewer #2: Yes

Reviewer #3: Yes

Reviewer #4: No

Reviewer #5: Yes

5. Review Comments to the Author

Reviewer #1: After reviewing this manuscript entitled: “Nonlinear Molecular Dynamics of quercetin: Its mechanistic role in hepatoprotection”. I consider that this article is very interesting but I have some concerns which could be improved in order to generate a major impact in readers:

1. In your study, you include an experimental model of hepatoprotection administrating two extracts. Thereby, I consider that your title should be modify as follows:

• Nonlinear Molecular Dynamics of Quercetin determined in Gynocardia odorata and Diospyros malabarica fruits: Its mechanistic role in hepatoprotection.

• However, you could include a similar title that highlights the two fruits involved in your study.

2. Please, include the main objective in your abstract.

3. In page 9 (line 60-61): Scientific names must be italicized. Please, check the whole manuscript.

4. Page 9; Lines 62-64: Authors refers silymarin, which is not the main metabolite of interest in this study. Please, verify your references and only explain the role of quercetin in hepatoprotection.

5. In regard to your introduction: I suggest the followings.

• Try to include only medicinal plants which quercetin has been the responsible effect in the hepatoprotective effect.

• Another paragraph explaining the mechanism of quercetin on the main targets involved in hepatoprotections with updated references.

• Include the main objective in the final paragraph of your introduction and/or secondary objectives.

In material and methods:

1. This sentence should be excluded of the Chemicals section (Molecular interaction and Molecular Dynamics studies were carried out in HP Workstation having (core i7, 3.9 GHZ 85 processor), 32GB RAM, 2TB HDD, NVIDIA Geforce GTX 1650ti graphics processor.)

2. In collection plant: include the GPS data, period of collection, months, where were fruits identified? Any herbarium.

3. In animals’ section: please, include the ethical approval of your institutional committee and reference any international guide for use of experimental animals.

4. In animals’ section: Try to improve your experimental design, how many males and females per group?

5. Type of food or balanced diet?

6. I cannot observe the methodology of the antioxidant activity in your main file. I consider that both antioxidant and histopathological studies should be included in the methodology and referenced.

7. For your antioxidant activity, you could use this reference: Hossain M. S, Uddin M. S, Kabir M. T, Begum M. M, Koushal P, Herrera-Calderon O, Akter R, Asaduzzaman M, Abdel-Daim M. M. In Vitro Screening for Phytochemicals and Antioxidant Activities of Syngonium Podophyllum L.: an Incredible Therapeutic Plant. Biomed Pharmacol J 2017;10(3). https://dx.doi.org/10.13005/bpj/1229

In your results, I cannot observe the table of liver function parameters and biochemical analysis (Table 2) in the main file and supplementary material.

The docking and dynamic analysis is well structured and written.

Your discussion is well planned.

General comments: Authors must correct my comments to improve some aspects such as: Order your methodology. Correct scientific names. Order your figures and tables according to your results. Please, verify and correct the references according to Plos One guide.

Reviewer #2: The manuscript entitled “Nonlinear Molecular Dynamics of quercetin: Its mechanistic role in hepatoprotection”

The liver performs number critical functions in the body. Accumulation of free radicals in liver may eventually cause damage, fibrosis, chirrhosis and cancer. Carbon tetrachloride (CCl4) belongs to hepatotoxin is converted to a highly reactive free radical by cytochrome P450 enzymes that causes liver damage. Plant extracts derived quercetin has substantial role in hepatoprotection. HPLC analysis revealed the abundance of quercetin in the fruit extracts of Gynocardia odorata and Diospyros malabarica, were isolated, purified and subjected to liver function analysis on Wistar rats. Post quercetin treatment improved liver function parameters in the hepatotoxic Wistar rats by augmenting bilirubin content, SGOT and SGPT activity. Gene expression profile of quercetin treated rats revealed down regulation of HGF, TIMP1and MMP2 expressed during CCl4 induction. In silico molecular mechanism prediction suggested that quercetin has a high affinity for cell signaling pathway proteins BCL, JAK and Cytochrome P450 CYP2E1, which all play a significant role in CCl4 induced

hepatotoxicity. In silico molecular docking and molecular dynamics simulation have shown that quercetin has a plausible affinity for major signaling proteins in liver. MMGBSA studies have revealed high binding of quercetin (ΔG) -41.48±11.02, -43.53±6.55 and -39.89±5.78 kcal/mol, with BCL-2 , JAK2 and Cyp2E1, respectively which led to better stability of the quercetin bound protein complexes. Therefore, quercetin can act as potent inhibitor against CCl4 induced hepatic injury by regulating BCL, JAK and Cyp2E1.

Manuscript written very well and extensive studies were done. Instead of having positive points I have seen few silly points which should be rectified before been accepted for publication. My review comments are provided below

1. BCL-2 and JAK2 naming are not uniform throughout the manuscript, uniformity should be maintained throughout the manuscript

2. CCL4, 4 should be in subscript

3. Line 82-83 Himedia India Pvt. Ltd should be incorporated

4. Line 167 Cyp2E1 should be CYP2E1, uniformity must be maintained.

5. Line 241-242 G.odorata and D. malabarica should be in italics as well as in line 248

6. After minute observation I have found in many places few words became conjugated together and that create problematic to read e.g. line 258 “D. malabaricaand” throughout the manuscript. Must be reviewed thoroughly and rectify accordingly.

Reviewer #3: The manuscript entitled “Nonlinear Molecular Dynamics of quercetin: Its mechanistic role in hepatoprotection” has engrossed extensive work and well planned manuscript. The results are presented well and authors have well executed each result. Apart from many pros I have seen some major concern which should be rectified or more deeply explained for better understanding. The manuscript can be accepted after the proper answers of the major and minor comments provided by the author. The comments are as follows:

Major Comments:

1. In, molecular docking studies, the major methodological segment must include the population size, selection of best pose based on RMSD clustering and the RMSD tolerance values. The rationale of selecting best dock poses having lowest binding energy score must be mentioned.

Minor Comments:

1. Beginning from abstract, BCL-2 and JAK2 nomenclature have discrepancies throughout the manuscript, somewhere BCL and JAK, in somewhere BCL-2 and JAK2 or in italics, uniformity should be maintained throughout the manuscript.

2. Line 202, MMGBSA equation should be properly aligned in single line

3. In line 241-242, 248 scientific names should be in italics and uniformity should maintain throughout the manuscript.

Reviewer #4: In this manuscript, Ghosh et al has studied the hepatoprotective potential of the Gynocardia odorata and Diospyros malabarica plant extracts derived quercetin. Authors have used qRT-PCR, in silico molecular docking, molecular dynamics simulation and histological methods to elaborate the hepatoprotection potential of the quercetin. However, there are some dots which need to be connected to make the study cleaner and clearer.

1- Line 19- # contributed equally, must be clear with whom author has contributed equally.

2- Line 20 in Abstract- The liver performs number critical functions in the body. Authors should correct this very first line of the manuscript.

3- There are many mistakes like in line 105- (SYS-LC-138, Systronics, India)with the mobile phase. Authors should revise the manuscript extensively to correct the syntax and grammar.

4- In experimental design, on page 11. “D. malabarica (body weight 200 mg/kg), G. odorata (body weight 200 mg/kg), and Silymarin (200 mg/kg body weight) were given orally to these groups of animals at 2, 24 and 48 h interval after the

administration of the last dose of CCl4 (Table 1)”. Authors must ensure that these intermittent doses must fall in the range of human effective dose to follow terms of translational medicine.

5- Quercetin, is very well established for its osteogenic role. Therefore, authors must ensure that there are not any heterotopic ossification or calcification concern, with the administration of the plants extracts, in the vital organs (heart and kidney).

6- In the Fig S2, Authors must keep the scale bar and it would better if authors replace the figures with higher magnification or more clear ones, to see better at cellular morphology.

7- Authors have used frozen liver tissue to isolate the RNA and perform the qRT-PCR. Authors must show the quantification and quality of the isolated RNA.

Reviewer #5: Dear Author

Thanks for the efforts that are put in this work on Nonlinear Molecular Dynamics of quercetin: Its mechanistic role in hepatoprotection. This work is a combination of in vivo and in silico CADD which is a welcome development in drug discovery process. The study has revealed that quercetin can act as potent inhibitor against CCl4 induced hepatic injury by regulating BCL, JAK and Cyp2E confirming it antioxidant potentials as a flavonoid

1. There are few issues that require revision:

Line 20 need to be revised to read; the liver performs a number of critical functions in the body

2. Line 21 and 22 need to also be revised for proper comprehension

1. How was the purity of the extracted quercetin measured?

2. Line 147 should be clarified. The authors reported molecular redocking, but there was no record of any docking in previous sections

3. Line 148-154 is not necessary, the commentary should be taken to discussion section

4. The diagrams are eye catching and interesting

5. Over all, it was a great study and should be accepted after minor revisions are made on the manuscript

6. PLOS authors have the option to publish the peer review history of their article (what does this mean?). If published, this will include your full peer review and any attached files.

Reviewer #1: No

Reviewer #2: No

Reviewer #3: **Yes: **MD BASHIR UDDIN

Reviewer #4: No

Reviewer #5: **Yes: **Daniel Ejim Uti PhD. Department of Biochemistry, Faculty of Basic Medical Sciences, College of Medical Sciences, Federal University of Health Sciences, Otukpo, Benue State Nigeria

---

## [Author Response · Author response to Decision Letter 0]

7 Jan 2022

Response to Reviewers

Review Comments to the Author

Reviewer #1

: After reviewing this manuscript entitled: “Nonlinear Molecular Dynamics of quercetin: Its mechanistic role in hepatoprotection”. I consider that this article is very interesting but I have some concerns which could be improved in order to generate a major impact in readers:

1. In your study, you include an experimental model of hepatoprotection administrating two extracts. Thereby, I consider that your title should be modify as follows:

• Nonlinear Molecular Dynamics of Quercetin determined in Gynocardia odorata and Diospyros malabarica fruits: Its mechanistic role in hepatoprotection.

• However, you could include a similar title that highlights the two fruits involved in your study.

Ans: We are thankful to reviewer’s concern and accordingly we have modified the title of the manuscript as suggested.

2. Please, include the main objective in your abstract.

Ans: The principal objective of our study has been incorporated in the abstract segment as “This study highlights the possible mechanism by which quercetin plays significant role in hepatoprotection” in the revised manuscript. 

3. In page 9 (line 60-61): Scientific names must be italicized. Please, check the whole manuscript.

Ans: We have rectified the scientific name as per the suggestion in the revised manuscript.

4. Page 9; Lines 62-64: Authors refers silymarin, which is not the main metabolite of interest in this study. Please, verify your references and only explain the role of quercetin in hepatoprotection.

Ans: We are thankful for the suggestion. But we would like to inform that the silymarin was taken as positive control/reference compound in our study which is well established for its hepatoprotection activity and therefore, have referred in the introduction section for better understanding.

5. In regard to your introduction: I suggest the followings.

• Try to include only medicinal plants which quercetin has been the responsible effect in the hepatoprotective effect.

• Another paragraph explaining the mechanism of quercetin on the main targets involved in hepatoprotections with updated references.

• Include the main objective in the final paragraph of your introduction and/or secondary objectives.

Ans: Revisions have been incorporated in the manuscript as:

• The plants reported in the introduction and discussion sections are rich source of quercetin and many of them were already reported elsewhere for hepatoprotective activity.

• As per the suggestion, mechanism has been displayed in the separate paragraph.

• Main objective of the study has been highlighted in the revised manuscript as per the suggestion.

In material and methods:

1. This sentence should be excluded of the Chemicals section (Molecular interaction and Molecular Dynamics studies were carried out in HP Workstation having (core i7, 3.9 GHZ 85 processor), 32GB RAM, 2TB HDD, NVIDIA Geforce GTX 1650ti graphics processor.)

Ans: As per the reviewers suggestion details of the computational infrastructure has been removed from the materials section in the revised manuscript. 

2. In collection plant: include the GPS data, period of collection, months, where were fruits identified? Any herbarium.

Ans: GPS location data has been incorporated in the revised manuscript as per suggestion. The collected plants were subjected to harbarium preparation and followed by identification at GUBH, the Dept. of Botany, Gauhati University, Assam, India. Nevertheless, accession numbers were assigned to the individual plants as D. malabarica (Acc. No. 18071 dt.04.11.2015) and G. odorata (Ac.No. 18072dt.04.11.2015). These statements have been incorporated in the revised manuscript.

3. In animals’ section: please, include the ethical approval of your institutional committee and reference any international guide for use of experimental animals.

Ans: Ethical approval statements have already been included in the previous version of the manuscript in the ethical approval section.

4. In animals’ section: Try to improve your experimental design, how many males and females per group?

Ans: In animals section, experimental setup has been improved for better understanding. In each group of 6 rats, 3 male and 3 females were taken and accordingly revised in the manuscript.

5. Type of food or balanced diet?

Ans: “Animal models were fed with Normal pellet diet” and has been incorporated in the revised manuscript. 

6. I cannot observe the methodology of the antioxidant activity in your main file. I consider that both antioxidant and histopathological studies should be included in the methodology and referenced.

Ans: The methodology of antioxidant activity and histopathological studies already incorporated in the supplementary file and a statement referencing that has been incorporated in the revised version.

7. For your antioxidant activity, you could use this reference: Hossain M. S, Uddin M. S, Kabir M. T, Begum M. M, Koushal P, Herrera-Calderon O, Akter R, Asaduzzaman M, Abdel-Daim M. M. In Vitro Screening for Phytochemicals and Antioxidant Activities of Syngonium Podophyllum L.: an Incredible Therapeutic Plant. Biomed Pharmacol J 2017;10(3). https://dx.doi.org/10.13005/bpj/1229

Ans: Thank you for your suggestion. We have incorporated this reference in the discussion as reference No.41. 

In your results, I cannot observe the table of liver function parameters and biochemical analysis (Table 2) in the main file and supplementary material.

Ans: The table of liver function parameters was shown in Table 2 in the original manuscript. The Biochemical analysis was shown in TableST2 in the supplementary file. 

The docking and dynamic analysis is well structured and written.

Your discussion is well planned.

Ans: We are grateful to the reviewer for appreciating our work.

General comments: Authors must correct my comments to improve some aspects such as: Order your methodology. Correct scientific names. Order your figures and tables according to your results. Please, verify and correct the references according to Plos One guide.

Ans: We have done modification to our revised version of the manuscript as suggested.

Reviewer #2: 

The manuscript entitled “Nonlinear Molecular Dynamics of quercetin: Its mechanistic role in hepatoprotection”

The liver performs number critical functions in the body. Accumulation of free radicals in liver may eventually cause damage, fibrosis, chirrhosis and cancer. Carbon tetrachloride (CCl4) belongs to hepatotoxin is converted to a highly reactive free radical by cytochrome P450 enzymes that causes liver damage. Plant extracts derived quercetin has substantial role in hepatoprotection. HPLC analysis revealed the abundance of quercetin in the fruit extracts of Gynocardia odorata and Diospyros malabarica, were isolated, purified and subjected to liver function analysis on Wistar rats. Post quercetin treatment improved liver function parameters in the hepatotoxic Wistar rats by augmenting bilirubin content, SGOT and SGPT activity. Gene expression profile of quercetin treated rats revealed down regulation of HGF, TIMP1and MMP2 expressed during CCl4 induction. In silico molecular mechanism prediction suggested that quercetin has a high affinity for cell signaling pathway proteins BCL, JAK and Cytochrome P450 CYP2E1, which all play a significant role in CCl4 induced

hepatotoxicity. In silico molecular docking and molecular dynamics simulation have shown that quercetin has a plausible affinity for major signaling proteins in liver. MMGBSA studies have revealed high binding of quercetin (ΔG) -41.48±11.02, -43.53±6.55 and -39.89±5.78 kcal/mol, with BCL-2 , JAK2 and Cyp2E1, respectively which led to better stability of the quercetin bound protein complexes. Therefore, quercetin can act as potent inhibitor against CCl4 induced hepatic injury by regulating BCL, JAK and Cyp2E1.

Manuscript written very well and extensive studies were done. Instead of having positive points I have seen few silly points which should be rectified before been accepted for publication. My review comments are provided below

1. BCL-2 and JAK2 naming are not uniform throughout the manuscript, uniformity should be maintained throughout the manuscript.

Ans: To maintain the uniformity we have done the rectifications throughout the manuscript.

2. CCL4, 4 should be in subscript

Ans: All CCL4 are subscripted in the revised manuscript as per the reviewers suggestion.

3. Line 82-83 Himedia India Pvt. Ltd should be incorporated

Ans: Incorporation has been done as per the reviewer’s suggestion.

4. Line 167 Cyp2E1 should be CYP2E1, uniformity must be maintained.

Ans: Uniformity of Cyp2E1 is maintained throughout the revised manuscript as per the reviewer’s suggestion.

5. Line 241-242 G.odorata and D. malabarica should be in italics as well as in line 248

Ans: Italicizing of the scientific names of plants G.odorata and D. malabarica are made in the revised manuscript

6. After minute observation I have found in many places few words became conjugated together and that create problematic to read e.g. line 258 “D. malabaricaand” throughout the manuscript. Must be reviewed thoroughly and rectify accordingly.

Ans: Changes have been made in the revised manuscript as per the suggestion.

Reviewer #3: 

The manuscript entitled “Nonlinear Molecular Dynamics of quercetin: Its mechanistic role in hepatoprotection” has engrossed extensive work and well planned manuscript. The results are presented well and authors have well executed each result. Apart from many pros I have seen some major concern which should be rectified or more deeply explained for better understanding. The manuscript can be accepted after the proper answers of the major and minor comments provided by the author. The comments are as follows:

Major Comments:

1. In, molecular docking studies, the major methodological segment must include the population size, selection of best pose based on RMSD clustering and the RMSD tolerance values. The rationale of selecting best dock poses having lowest binding energy score must be mentioned.

Ans: In methodological segments statement has been introduced as “During molecular docking studies, three replicates were performed having the total number of solutions computed 50 in each case, with population size 500, number of evaluations 2500000, maximum number of generations 27000 and rest the default parameters were allowed. After docking, the RMSD clustering maps were obtained by re-clustering with a clustering tolerance 0.25 Å, 0.5 Å and 1 Å, respectively, in order to obtain the best cluster having lowest energy score with high number of populations.” in the revised mansucript as per the suggestion

In results section a new statement has been incorporated in the revised manuscript results section of molecular dcoking for better understanding of the rationale of selection of best dock pose as “The best dock pose was seleccted based on low RMSD tolerance 0.5 Å and binding energy having maximum within that RMSD cluster” 

Minor Comments:

1. Beginning from abstract, BCL-2 and JAK2 nomenclature have discrepancies throughout the manuscript, somewhere BCL and JAK, in somewhere BCL-2 and JAK2 or in italics, uniformity should be maintained throughout the manuscript.

Ans: All rectifications are done in the revised manuscript as per the suggestion.

2. Line 202, MMGBSA equation should be properly aligned in single line

Ans: MMGBSA equation is properly aligned in a single line in the revised manuscript.

3. In line 241-242, 248 scientific names should be in italics and uniformity should maintain throughout the manuscript.

Ans: Scientific names are italicized at the respective positions throughout the revised manuscript as per the suggestion

Reviewer #4: 

In this manuscript, Ghosh et al has studied the hepatoprotective potential of the Gynocardia odorata and Diospyros malabarica plant extracts derived quercetin. Authors have used qRT-PCR, in silico molecular docking, molecular dynamics simulation and histological methods to elaborate the hepatoprotection potential of the quercetin. However, there are some dots which need to be connected to make the study cleaner and clearer.

1- Line 19- # contributed equally, must be clear with whom author has contributed equally.

Ans: As per the reviewer’s suggestion equal contribution is marked “†” to Arabinda Ghosh and Pranjal Sarmah in the revised mascript for better undersanding.

2- Line 20 in Abstract- The liver performs number critical functions in the body. Authors should correct this very first line of the manuscript.

Ans: This statement has been modified as per the reviewer’s suggestion in the revised manuscript.

3- There are many mistakes like in line 105- (SYS-LC-138, Systronics, India) with the mobile phase. Authors should revise the manuscript extensively to correct the syntax and grammar.

Ans: The statement and the grammar has been thoroughly checked and rectified in the revised manuscript.

4- In experimental design, on page 11. “D. malabarica (body weight 200 mg/kg), G. odorata (body weight 200 mg/kg), and Silymarin (200 mg/kg body weight) were given orally to these groups of animals at 2, 24 and 48 h interval after the

administration of the last dose of CCl4 (Table 1)”. Authors must ensure that these intermittent doses must fall in the range of human effective dose to follow terms of translational medicine.

Ans: We have followed the OECD(423) guideline while designing the experiment as mentioned in the methodology section of our manuscript. The dose selection was done by strictly following the guideline. Further, we found that the dose falls in the range of Human Equivalent Dose (HED) based on the following standard conversion formula :

HED (mg/Kg) = Animal Dose(mg/Kg) x (Animal Km/Human Km) [Nair and Jacob, 2016; Journal Basic Clinical pharmacy]

5- Quercetin, is very well established for its osteogenic role. Therefore, authors must ensure that there are not any heterotopic ossification or calcification concern, with the administration of the plants extracts, in the vital organs (heart and kidney).

Ans: We appreciate the reviewer for his/her concern. However, as our work focuses the mechanistic role of quercetin in hepatoprotection, so, we have not included the consequent impact of these plant extracts on other organs in the present study.

On the other hand, the most recent report revealed that the quercetin helps in preventing heterotopic ossification (Li et al, 2021; Frontiers in Immunology Vol:2; doi: 10.3389/fimmu.2021.649285). Similarly, it was also reported that quercetin significantly lower the severity of ossification (Chang et al . 2017; BioMed Res Int.; doi: 1155/2017/5716204).

6- In the Fig S2, Authors must keep the scale bar and it would better if authors replace the figures with higher magnification or more clear ones, to see better at cellular morphology.

Ans: The Fig S2 has been modified as per the suggestion of the Reviewer and incorporated in the revised supplementary file S1.

7- Authors have used frozen liver tissue to isolate the RNA and perform the qRT-PCR. Authors must show the quantification and quality of the isolated RNA.

Ans: We have incorporated the quantification and quality of the isolated RNA in the revised version of the manuscript and the data were shown as supplementary file.

 The 260/280 ratio for the RNA isolated from the liver tissue samples ranged from 2.08-2.14 suggesting good quality RNA (Supplementary File S2, Table ST3). The integrity of RNA was checked on Agarose gel showing discrete 28S and 18S ribosomal RNA band on each sample suggesting that the RNA in each case was intact and could be used for qPCR analysis (Supplementary File S2, Fig S3).

Reviewer #5: 

Dear Author

Thanks for the efforts that are put in this work on Nonlinear Molecular Dynamics of quercetin: Its mechanistic role in hepatoprotection. This work is a combination of in vivo and in silico CADD which is a welcome development in drug discovery process. The study has revealed that quercetin can act as potent inhibitor against CCl4 induced hepatic injury by regulating BCL, JAK and Cyp2E confirming it antioxidant potentials as a flavonoid

1. There are few issues that require revision: Line 20 need to be revised to read; the liver performs a number of critical functions in the body

Ans: The statement has been revised in the manuscript as “Liver performs number of critical physiological functions in human system”

2. Line 21 and 22 need to also be revised for proper comprehension

Ans: Statement has been revised as “Intoxication of liver leads to accumulation of free radicals that eventually cause damage, fibrosis, cirrhosis and cancer.”

1. How was the purity of the extracted quercetin measured?

Ans: Purity of the extracted quercetin from fruit extracts were determined by comparing with the standard Quercetin in HPLC. 

2. Line 147 should be clarified. The authors reported molecular redocking, but there was no record of any docking in previous sections

Ans: This statement meant the docking performed first using Autodock 4.2 and then performed in Schrodinger 2018-4 using glide and QM docking. Details of docking already mentioned in methods section.

3. Line 148-154 is not necessary, the commentary should be taken to discussion section

Ans: The commentary has been removed from methods section as per the reviewer’s suggestion, and similar statements were already reported in the discussion section in the previous version of the manuscript.

4. The diagrams are eye catching and interesting

Ans: We are thankful to the reviewer for appreciating our work.

5. Over all, it was a great study and should be accepted after minor revisions are made on the manuscript

Ans: We are grateful to the reviewer for appreciating our work.

---

## [Decision Letter · Decision Letter 1]

31 Jan 2022

Nonlinear Molecular Dynamics of Quercetin in Gynocardia odorata and Diospyros malabarica fruits : Its mechanistic role in hepatoprotection

PONE-D-21-29456R1

Dear Dr. Baishya,

We’re pleased to inform you that your manuscript has been judged scientifically suitable for publication and will be formally accepted for publication once it meets all outstanding technical requirements.

Kind regards,

Ghulam Md Ashraf, Ph.D.

Academic Editor

PLOS ONE

Additional Editor Comments (optional):

The authors have addressed all the comments and the manuscript is now acceptable for publication.

Reviewers' comments:

Reviewer's Responses to Questions

**Comments to the Author**

1. If the authors have adequately addressed your comments raised in a previous round of review and you feel that this manuscript is now acceptable for publication, you may indicate that here to bypass the “Comments to the Author” section, enter your conflict of interest statement in the “Confidential to Editor” section, and submit your "Accept" recommendation.

Reviewer #1: All comments have been addressed

Reviewer #2: All comments have been addressed

Reviewer #3: All comments have been addressed

Reviewer #4: All comments have been addressed

Reviewer #5: All comments have been addressed

2. Is the manuscript technically sound, and do the data support the conclusions?

Reviewer #1: Yes

Reviewer #2: Yes

Reviewer #3: Yes

Reviewer #4: Yes

Reviewer #5: Yes

3. Has the statistical analysis been performed appropriately and rigorously? 

Reviewer #1: Yes

Reviewer #2: Yes

Reviewer #3: Yes

Reviewer #4: Yes

Reviewer #5: Yes

4. Have the authors made all data underlying the findings in their manuscript fully available?

Reviewer #1: Yes

Reviewer #2: Yes

Reviewer #3: Yes

Reviewer #4: Yes

Reviewer #5: Yes

5. Is the manuscript presented in an intelligible fashion and written in standard English?

Reviewer #1: Yes

Reviewer #2: Yes

Reviewer #3: Yes

Reviewer #4: Yes

Reviewer #5: Yes

6. Review Comments to the Author

Reviewer #1: The manuscript entitled "Nonlinear Molecular Dynamics of Quercetin in Gynocardia odorata and Diospyros

malabarica fruits : Its mechanistic role in hepatoprotection" has been improved according to my comments and could be accepted in its current form.

Reviewer #2: All comments were properly addressed and clarified. So, the manuscript may be accepted in this Journal

Reviewer #3: All comments are well addressed and could be Accepted for publication. Editing manuscript Language also well improved.

Reviewer #4: Ghosh et. al, have beautifuly addressed, point by point, all my comments and corrected the manuscript accordingly.

Reviewer #5: Dear Author

Thanks for the efforts that are put in this work on Nonlinear Molecular Dynamics of quercetin: Its mechanistic role in hepatoprotection. This work is a combination of in vivo and in silico CADD which is a welcome development in drug discovery process. The study has revealed that quercetin can act as potent inhibitor against CCl4 induced hepatic injury by regulating BCL, JAK and Cyp2E confirming it antioxidant potentials as a flavonoid. The minor issues raised in the earlier manuscript were well addressed in this revised copy. Over all, it was a great study and should be accepted for publication

7. PLOS authors have the option to publish the peer review history of their article (what does this mean?). If published, this will include your full peer review and any attached files.

Reviewer #1: No

Reviewer #2: No

Reviewer #3: No

Reviewer #4: No

Reviewer #5: **Yes: **Daniel Ejim Uti (PhD.)

---

## [Editor Report · Acceptance letter]

10 Feb 2022

PONE-D-21-29456R1 

Nonlinear Molecular Dynamics of Quercetin in *Gynocardia odorata* and *Diospyros malabarica* fruits : Its mechanistic role in hepatoprotection 

Dear Dr. Baishya:

I'm pleased to inform you that your manuscript has been deemed suitable for publication in PLOS ONE. Congratulations! Your manuscript is now with our production department. 

Kind regards, 

on behalf of

Dr. Ghulam Md Ashraf 

Academic Editor

PLOS ONE